# Mental health and cultural and linguistic diversity as challenges in school? An interview study on the implications for students and teachers

**Sanna Higgen**[1]*, **Mike Mösko**[1,2]

**1** Department of Medical Psychology, University Medical Center Hamburg-Eppendorf, Hamburg, Germany,
**2** Department of Applied Human Sciences, Magdeburg-Stendal University of Applied Sciences, Stendal, Germany

* s.higgen@uke.de

## Abstract

Mental health and cultural and linguistic diversity in classrooms are part of students' lives. Both factors can lower the achievements of students and classrooms and pose a challenge for teachers. Yet, little is known about the effects on other areas of school life besides achievements. Also, the consequences for classmates and teachers as well as possible resources are mostly disregarded. Semi-structured interviews were held to investigate the diverse effects of mental health issues and cultural and linguistic diversity on students, classmates and teachers. In total 20 interviews were conducted, seven with teachers, seven with external professionals and six with students. Recordings were transcribed and analysed using qualitative text analysis. Results show that especially externalizing symptoms of mental health issues are a burden to classmates and teachers. Teachers face time management problems and emotional stress. Linguistic diversity constitutes a serious challenge at school for students, classmates and teachers. Yet, cultural and linguistic diversity also imports several resources like fostering openness and integration. Future research should expand this research to older children and investigate the specific needs of teachers.

**Data Availability Statement:** All relevant data are within the paper and its Supporting Information files.

## Introduction

Worldwide 13.4% of children suffer from a mental disorder [1]. Around one-third of youth experience some form of disorder during their lifetime [2]. If subthreshold symptoms are considered as well, approximately half of the adolescents across eleven countries meet criteria for threshold and/or subthreshold-depression and/or anxiety [3]. As schools are the place where children spend a great amount of their time, the effects of their mental health problems naturally impact the school environment [4–7]. Another factor that shapes this environment is the cultural and linguistic diversity of students [8–11]. The number of international migrants is continuing to increase in recent years [12, 13]. 13.9% of these are below the age of 20 [13]. In Germany, around 10.7% of all students are foreign students that cannot be included in regular

**Funding:** The author was supported by the Hans-Böckler-Foundation through a doctoral scholarship. https://www.boeckler.de/index.htm. There was no additional funding for this project.

**Competing interests:** The authors have declared that no competing interests exist.

classrooms [14]. When looking at the migration background the numbers are even higher [15]. Mental health problems as well as cultural and linguistic diversity both contribute to students' diversity and impact the school experiences of the children themselves, the classmates as well as the teachers.

## Mental health

Psychological stress and mental health disorders can negatively affect the achievements in school, the educational career [4, 5, 16] and the cognitive abilities of the affected children [17]. Studies on Canadian and American children show that this in turn, can lead to grade repetition [17] and leaving school prematurely, raising the risk of unemployment and illiteracy [18]. Furthermore, mental health problems are associated with lower health-related quality of life [19], difficulties in relationships to peers and teachers [6], an increased likelihood of being victimised or bullied by peers [7, 20, 21] and peer rejection [22, 23].

In addition to these negative impacts of mental health problems, classmates of distressed students are affected as well. Student misbehaviour can deteriorate the classroom environment [24] which is related to students' life satisfaction according to a German study [25]. Some research shows that including a disruptive pupil into the school-cohort lowers the academic achievements of their classmates [26–28]. In a US sample emotionally disabled children decreased the reading and maths grades of their peers by over 10 percent of a standard deviation [27]. Besides, teachers spend a great amount of time on disciplining disruptive students [29] which diminishes learning time for the rest of the class.

The majority of teachers has had students with mental health problems in their classes [7, 30] which shows that it is a prevalent issue in school. Yet, many teachers feel isolated [31], incompetent in supporting these children [30, 32] and lack professional training in dealing with mental health issues [7]. The mental health needs of their students pose a burden on the teachers [33]. A qualitative study with teachers from Australia investigated the teachers' perspective on externalising and internalising problems in high school students. The teachers identified children with externalising symptoms as being more disruptive which can impact the entire classroom [34] and cause teachers' stress [34, 35]. Also, troublesome students can deteriorate the teachers' perception of the classroom environment [36, 37].

## Cultural and linguistic diversity

Another important factor that affects the classroom environment is the cultural and linguistic background of students. A Germany study found that migration background is associated with lower achievements in students [8]. Language deficiencies form a barrier to higher education, which are largely determined by the language spoken at home [8]. An international meta-analysis conducted by van Ewijk and Sleegers [9] concluded that a higher share of students from one ethnic minority affects pupils' achievements of the same minority. If students are used to a more traditional classroom environment they might not participate as much in an intercultural classroom [38]. Besides, in a Belgium sample non-native students were at a higher risk of being victimised, especially in schools where they constituted a minority [39].

In some cases, cultural and linguistic diversity leads to decreased achievements of the entire classroom [8, 40, 41]. However, this effect is controversial and was attributed to socio-economic status in other studies [8, 42–44]. Goldsmith [10] and Moody [45] detected that with increasing ethnic heterogeneity at schools in the United States, levels of interethnic conflict and friendship segregation were augmented. A side effect of ethnic heterogeneity in Flemish schools was an increase in a feeling of futility, which lead to a rise in school misconduct by native students [46]. Migrant children, however, seemed to be unaffected by the ethnic

composition of students at least when looking at misconduct [46]. A review by Gray-Little and Hafdahl [47] found that with increasing consonance of a racial group in a school setting the self-esteem of that particular group amplifies accordingly.

The cultural and linguistic background of students can even alter the teaching style. By adapting their lessons to the language deficiencies of students, teachers can inadvertently be responsible for lower achievements in a classroom [8]. Besides, differing perceptions of race by teachers can lead to negative expectations of minority students and impact on the students' achievements as was shown in studies from the United States [48, 49]. Teachers face the challenge of having to recognise differences between students while avoiding assimilation or essentialism [50, 51]. They are required to possess a range of teaching styles to meet the variety of needs of their students [52–54]. Flores and Smith [11] suggest that teachers need diversity preparation to be able to create an equitable learning environment. They need to go through the difficult process of reflecting their own feelings, their own race, and attitudes about race to provide children with an education that is free from prejudice [52, 55, 56]. To be able to teach diverse classes effectively teachers need to develop intercultural competence and simultaneously foster this competence in students [52, 57, 58]. Intercultural competence is the capability of changing the cultural perspective and adapting to cultural differences [59, 60]. It is the competence to encounter other cultures positively [51] and to interpret oneself in varying cultural ways [60]. For teachers, intercultural competence refers to the ability to think and act in interculturally appropriate ways in the context of the school and classrooms [61, 62]. Intercultural competence in teachers can be increased through training cultural self-reflection and awareness [63, 64].

Most investigations on cultural and linguistic diverse classrooms have focused on the negative impacts on students and teachers. However, as literature from the United States from other contexts than school depicts, multiculturalism can have positive impacts like increasing a firm's performance [65, 66] or enriching the education of college [67] and medical students [68]. In the school context, Stanat, Schwippert [8] found, that the share of students with migration background positively influences the aspiration for higher education in students. Besides, a greater diversity of a high school student's environment in the United States was associated with more comfort with classmates from different ethnic or racial groups [69].

The literature identifies some effects of mental health and cultural and linguistic diversity on students and teachers. Yet, most research has focused on the academic performance of students neglecting personal consequences for students and teachers or interactions between the varying parties. Besides, there is only little information on the possible resources found in culturally and linguistically diverse classrooms. Therefore, semi-structured interviews with teachers, students, and external professionals were conducted to investigate their perspective on the effects of mental health and cultural and linguistic diversity in school.

## Methods

### Ethics

Ethics approval was obtained from the ethics committee of the chamber of psychotherapists Hamburg on September 7th 2018 (application no. 04/2018-PTK.HH). The agency for school and vocational education approved of the study on October 11th 2018. At the beginning of each interview, participants received oral and written information about the purpose of the study and a description of the ethical rules used for research including confidentiality, informed consent, information and voluntary participation [70]. Compliance with the data privacy laws according to the §27 HmbDSG on the treatment of personal data for the purpose of scientific research was assured. All participants gave written informed consent for the interviews to be recorded, the recordings to be transcribed and the results of the study to be

published anonymously. Parental permission was obtained for all children that participated in these interviews.

## Researcher characteristics

The first author conducted and analysed all interviews. SH is a female master's in psychology and a doctoral researcher (exemplary research areas: resilience, primary school, prevention). She was a first-time interviewer but received training in conducting interviews and qualitative data analysis from the research group.

## Recruitment

Participants were selected based on a purposive sampling approach [71]. The recruitment of the primary school teachers aimed at maximum variation in respect to age, years of work experience and location of the school. Schools were chosen from three different locations: One is located in a small rural town with around 40% of students that have a migration background. The other two are in a large city. One is situated in a high-income suburb where only 5% of students have a migration background. The other one is in a low-income suburb where over 70% of students have a migration background. Seven teachers were recruited, three from the small town and four from the bigger city, two from each school. All had experience with culturally and linguistic diverse classes and mental health issues in students. The schools were contacted via mail or personal contact. Teachers that might be interested in participating in the interview were recommended by their colleagues or the head of school. The teachers were reached via telephone or mail. No teacher refused to participate in the interview.

Students were recruited through contacts made with the interviewed teachers. Students were supposed to have experience with mental health issues in classmates and diverse classrooms. Besides, they should variate in gender and age. No student refused to participate in the interview.

In addition, external professionals that work in areas related to mental health and cultural and linguistic diversity were interviewed. They were recruited as a key informant sample based on their experience. The research team discussed the choice of externals and agreed on the professions that should be interviewed. An online search was conducted to identify one special needs educator and the two psychologists. The other externals were recruited through contacts with experts and word-of-mouth. Interview partners were contacted via email. Two contacted externals did not agree to give an interview because of limited time resources or legal issues. Two did not reply to the emails they were sent.

## Data collection

Interviews were held between September and November 2018. They were based on a semi-structured guideline [72] and carried out at the participants' homes, their workplace, or school. Interviews were always conducted in a private one-on-one setting. Only one interview was conducted simultaneously with an external and a teacher because the external had invited the teacher without advanced notice to the interviewer. The interviews were digitally audio recorded. They lasted on average $M = 32:46$ (range = 22:19–50:13) minutes with teachers, $M = 31:41$ (range = 17:48–50:13) minutes with externals and $M = 22:21$ (range = 12:52–26:49) minutes with students.

## Guideline

The content of the interviews focused on experience with mental health issues in children and cultural and linguistic diverse classrooms. The guidelines were pilot tested with one teacher.

No changes were made to the guidelines following the pilot testing. The first part of the interviews with teachers and externals contained questions on mental health issues in children. They were asked how it might be expressed in school, what influences this expression can have on the class and the classmates and what challenges thereby arise for teachers. The second part of the interviews focused on cultural and linguistic diverse classrooms. Teachers and externals were asked what kind of challenges exist for the students and teachers and what resources can be found.

The guidelines for students were developed and discussed with the research team. They were piloted with one child. Subsequently, a few more specific questions were added to the guidelines. Students were asked what they do when they feel unwell, how their friends and teachers could help them, whether they noticed when a friend is feeling unwell and what they themselves or the teachers could do to help them.

## Protection of data privacy

In order to protect the participants' data privacy, informed consent was separated from the recorded data. All names and identifying information were replaced by labels and numbers during the transcription process.

## Transcription

Members of the research team transcribed the interviews verbatim. All transcripts were checked for accuracy by the main author. Transcripts were not returned to participants for correction.

## Data analysis

Transcripts were analysed according to the qualitative text analysis by Kuckartz [73] using Maxqda Analytics Pro 2010. In qualitative text analysis by Kuckartz [73] a structure is formed by formulating main categories or extracting these from the transcripts in a first coding process. The research team discusses the main categories and identifies subcategories. Subsequently, all interviews are coded again using the developed category system.

Deductive categories were formed based on the interview guideline. During the first coding process, other main categories emerged. Two members of the research team coded five interviews separately. They compared their deduced codes and discussed questions that arose during coding.

Codes could be anything between a few words or an entire paragraph. All codes were described in code memos and linked with a supporting quote. All transcripts were coded once. Then the entire material was coded again using the established code system. The final codes were discussed with the research team to identify appropriate labels.

## Results

### Sample

The sample consists of seven female primary school teachers, six children, and seven external professionals with specific knowledge on the matter. The teachers were on average $M = 47.86$ (range = 33–71) years old and had $M = 17.57$ (range = 2–42) years of teaching experience. One teacher was born in Russia and one in Brazil. The others were all German. Students were either seven ($n = 3$) or ten ($n = 3$) years old and were in first or fifth grade. In each grade two girls and one boy were interviewed. Externals were $M = 50.86$ (range = 35–69) years old with $M = 20$ (range = 5–43) years of work experience. There were two special needs educators

(male and female), two psychologists (male and female), one paediatrician (female), one worker from the German Child Protection Agency (female), and one social space manager (female).

## Coding

Regarding the question of the effects of mental health problems in students the main category "mental health problems" splits into the sub-codes "individuals", "classmates", "teachers" and "resources". Each sub-code consists of multiple subcategories (Table 1). Concerning the questions of the effects of cultural and linguistic diversity the main category "cultural and linguistic diversity" divides into the same sub-codes, namely "individuals", "classmates", "teachers" and "resources". These sub-codes also consist of multiple subcategories (Table 2).

Teachers and externals expressed similar views on the effects of mental health and cultural and linguistic diversity on students and teachers. Every subcategory was mentioned by both sides. However, teachers and externals occasionally mentioned different examples. In those cases where there are differences between both groups, they will be pointed out in the results. Students' comments mostly serve as an enrichment of the data.

## Mental health

**Individuals.** When looking at mental health problems of individuals in school, the most dominant issue in the interviews was the damaging effect of *externalising behaviour*. This behaviour causes students to disrupt lessons and act out against teachers. Many suffer from restlessness. The interviewed teachers also mentioned it causes problems during the break time when students have more freedom. Students and teachers named many instances, where children acted with aggression towards classmates and against themselves.

Mental health problems can also manifest themselves in *withdrawal*. Withdrawal implies no participation in school and little talking. Teachers, as well as externals, have even experienced children that dissociate or show selective mutism. These students do not interrupt the school lesson.

Yet, according to teachers and externals, both externalizing and internalising behaviour can cause *social problems* because children have difficulties finding friends. Especially conflict communication was named as challenging for a lot of them as they exhibit low frustration tolerance.

Besides these forms of expressing their problems, some students show *inappropriate behaviour*. One teacher mentioned a very problematic student who would steal things from classmates and teachers or imitate death by asphyxiation. Other teachers had students that showed infant-like behaviour or motoric difficulties. One student and two externals talked about children that would show or talk about sexual behaviour. Finally, the German Child Protection Agency worker had witnessed self-harming.

Mental health problems often go hand in hand with a feeling of *excessive demand* by everything that is asked of the children. The children feel stressed and sad. The exerted pressure on these children can make it impossible for them to focus on school requirements.

Consequently, mental health problems can lead to a *deteriorated school performance*. Many students have difficulties concentrating because their mind is full of other issues. According to two externals, some children show disinterest in school and even resort to avoidance of school, if the situation becomes too unbearable.

Mental health problems sometimes manifest in *somatic symptoms*. As the child clinician pointed out many children see a doctor for physical problems, which are frequently not recognised as symptoms of mental health problems in the first instance.

**Table 1. Effects of mental health problems in children.**

| Individuals | | |
|---|---|---|
| **Code** | **Example** | **Citation** |
| Externalising behaviour | Disrupt lessons | Teacher 4 *"They have to let their inner unwellness, the aggression, that they also have, out with movements, with verbal misfiring."* |
| | Restlessness | |
| | Aggression | |
| | Trouble during break time | |
| Withdrawal | Dissociate | External 6 *"We know children, who simply go silent, withdraw, who don't talk anymore, who don't want to visit the school as such anymore."* |
| | Mutism | |
| Social problems | Few friends | External 4 *"They have more trouble finding playmates. Depending on the psychological distress, it can be very very different. Also, a child that withdraws will eventually not be asked anymore whether it can join in or a child that acts peculiarly will actually often not be included in play situations."* |
| | Troubled conflict communication | |
| Inappropriate behaviour | Theft | Teacher 1 *"It shows in his behaviour, in the social context, many things that are simply very peculiar and that shouldn't be like that."* |
| | Motoric difficulties | |
| | Age-inadequate behaviour | |
| | Sexual behaviour | |
| | Self-harming | |
| Excessive demand | Stress | Teacher 5 *"I did tell the parents in a discussion with them, I feel like she can't take it anymore. She simply can't take it anymore. The pressure is so big for her, at a different level, so that the topics school and performance at school, they don't get through all these frosted glass tiles that eventually block the sight."* |
| | Sadness | |
| | Torn between teachers and parents | |
| Deteriorated school performance | Blocked Mind | Teacher 6 *"I had them [children with psychological distress] in my lessons and I asserted that they are very much occupied with other things, with themselves. And that they have difficulties finding access to deal with learning content because they are mentally not ready for that. Because they have other things that are much more dominant and formative for them in the school situation. Even though there they are not in the situation that causes the distress."* |
| | Difficulties concentrating | |
| | Avoidance of school | |
| | Disinterest | |
| Somatic symptoms | Stomach ache | External 5 *"The symptoms are manifold and are not always immediately interpreted as an expression of psychological distress. The children and that has been increasing over the past years, suffer from headaches, stomach aches without organic evidence, I did experience that a lot. The children suffer from sleep disturbances, poor appetite, also from symptoms that you first attribute to organic disease and only after exclusion of an organic disease you talk to the parents about what other possible causes might be considered."* |
| | Headache | |
| | Sleep disturbance | |
| | Loss of appetite | |
| Classmates | | |
| Devaluation | Attract attention | External 6 *"One child with some slight abnormality quickly becomes an object of special interest for students. And they can then partly also react derogative, partly exclude, partly be irritated, uncomprehending."* |
| | Stigmatisation | |
| | Exclusion | |
| | Bullying | |
| | Incomprehension | |
| Instigation | Participation | Student 3 *"I didn't want that but they said I should join in. And my friend was also pushed into it. And then he did it as well. That was a bit stupid."* |
| | Reciprocal effects | |
| Burden | Deteriorated classroom atmosphere | Student 2 *"Actually we were scared because it had happened to us, that we were treated like that. We didn't want to go to school anymore, because we were so scared that he would beat us up."* |
| | Fear | |
| | Lower achievements | |
| | Disturbance during lessons | |
| Teachers | | |
| Personal burden | Uncertain responsibilities | External 1 *"I go home and if I don't have a good environment then I carry it with me. It doesn't work any other way. I can't say for myself: Oh well, I have to change that so that it doesn't bother me". Because that is the problem. Eventually, you take it personally then it affects you and it eats away at you."* |
| | Psychological or emotional stress | |
| | Burn-out | |
| | Additional appointments | |
| | Feeling left alone | |
| | Feeling incompetent | |

*(Continued)*

**Table 1.** (Continued)

| Individuals | | |
|---|---|---|
| **Code** | **Example** | **Citation** |
| Time management | Following the curriculum | Teacher 4 *"Especially after breaks, when the children have more freedom, more than usual, where they let it out, you constantly have to reprocess it. You first have to talk about the conflicts before you can turn to the actual lesson."* |
| | Solving conflicts | |
| | Disruption of lessons | |
| Care for all students | Distribution of attention | External 6 *"[Teachers] often have the experience, that they don't do justice to the single child. Or the others if they dedicate themselves again too much to that child. So, this balancing act which isn't easy because of inclusion and other challenges, of course, becomes not easy through mental disorders."* |
| | Neglecting students | |
| | Intervening | |
| | Integrating all students | |
| | Protecting students | |
| | Training social competencies | |
| Skills | Self-reflection | External 7 *"You have to watch out that you treat all children equally. Not, you are loud, that's why we don't want you. They are different, they present themselves differently. One is silent, one is loud, one is agitated, another one is disinterested, one is diligent, one lazy."* |
| | Admitting to mistakes | |
| | Patience | |
| | Interpreting behaviour | |
| | Differentiated treatment | |
| Collaboration with parents | Missing transparency | Teacher 7 *"Working with the parents of distressed children is a big, time-intensive but very necessary additional task in the daily business. You can't catch a child without working with the parents. Establishing a trustful contact, reducing the fear of thresholds and then finding solutions for the child together with the parents, demands a lot of time and tact. Sometimes you have to make house calls, you invite the parents to lessons and you always have to lend an ear even without fixed appointments."* |
| | Challenging cooperation | |
| | Blaming culture | |
| Resources | | |
| Support | Recognise emotions | Teacher 1 *"The students, you notice that they are very very helpful even though he has already threatened many children."* |
| | Understanding | |
| | Helping behaviour | |
| | Distraction | |
| | Mediation | |
| Acceptance | Accepting differences Ignoring | External 2 *"There are students who handle it very relaxed, who say "well, so, there is someone, who is a bit different, but being different is completely fine" and there are also teachers who can handle it well."* |

The table presents the codes that were extracted from the interviews concerning the consequences of mental health problems in children for the children themselves, the classmates and the teachers. Every code is explained with examples as well as a citation.

**Classmates.** The peers of children with mental health issues show varying reactions, which form three categories. In some cases, the behaviour of distressed children causes others in the classroom to *devaluate* this child. Even little aberrances can call the attention of students and bring them to focus on these. Experts pointed out that the peers might stigmatise this child for example as "the clown". One psychologist mentioned that some mental disorders like obsessive-compulsive disorders or tic disorders can disrupt and irritate the entire class. Aggressive or aberrant behaviour can even lead to exclusion or bullying. Some externals stated that teachers might have a hard time explaining differential behaviour or treatment to the other children in the classroom.

Sometimes distressed children *instigate* classmates to participate in disrupting the lesson. One student talked about having been incited into bullying a teacher with another student. If peers suffer from a similar mental health problem, they often do not want one child to gather all the attention of the teachers and students. The reaction of the classroom can in turn influence the subsequent behaviour of the distressed child.

**Table 2. Effects of the cultural and linguistic diversity of students.**

| Individuals | | |
|---|---|---|
| **Code** | **Example** | **Citation** |
| Native language deficits | Learning difficulties | Teacher 3 *"Generally, it's the case that these children obviously can't participate in the regular lesson. They can't follow the lesson the way you would hope for those children and also for the entire classroom. It's the thing that they simply can't participate because of the language barrier also."* |
| | Concentration problems | |
| | Low self-esteem | |
| | Conflict communication | |
| Social affiliation | Home in multiple countries | External 7 *"That's why it's a challenge, especially when they [the students] come together. Because these children, they are simply sorted out. This person not and this person not and the teacher is German as well. Oh god, that means I am a stranger here."* |
| | Ethnic grouping | |
| | Few friends | |
| Difficult preconditions | Lower financial support | Teacher 1 *"Especially in this case, with this one student, you can see that the children that live in the refugee camp, of course, they notice that they have different conditions, also materialistically. That is obviously very contrary to this district, where the children receive everything threefold and the best and prettiest schoolbags and "Schultüten"* self-made and I don't know what and he doesn't have that, he only has the simplest means."* * cornet made of cardboard filled with sweets and presents given to children in Germany on their first day of school |
| | No schooling experience | |
| Strain through experiences | Posttraumatic stress disorders | External 3 *"And additionally, some children that experienced a flight, they have posttraumatic stress disorders or at least symptoms of it, often they find it particularly hard to concentrate."* |
| | Concentration problems | |
| | Violence as a solution | |
| **Classmates** | | |
| Incomprehension | Behaviour | Teacher 3 *"The challenges are of course that the children were very differently brought up maybe also culturally, that they might not be able to understand how a child behaves or how it might have been brought up at home. That can obviously also lead to misunderstandings, also the reaction of some children can lead to misunderstandings."* |
| | Food | |
| | Differing treatment | |
| Exclusion | Bullying | Teacher 3 *"This can also lead to segregation, that you, when you don't understand it, take distance to those children. And that can also lead to social problems, that the children are like I said excluded, that they don't have friends. I think that is a big risk here because you do isolate yourself a lot."* |
| | Distancing | |
| | Special treatment | |
| | Outsider | |
| Barriers through differences | Religion (Ramadan) | Teacher 6 *"I believe religion plays a major role. The kids absolutely don't care whether someone looks somehow Asian, African, Indian. But it is always this, maybe also especially in this district, will you celebrate Christmas? Or St. Nicolas doesn't exist."* |
| | Public holidays | |
| | Values | |
| Challenging expectations | Expected to be<br>• considerate<br>• independent<br>• patient | External 3 *"When for example a colleague has to explain some things over and over because it doesn't work on the language level, it gets boring for those who did already understand. Then one starts to tilt on his chair, then the next one starts throwing things. That is often difficult to stand. Or also for the other children when you explain a game during a sports lesson. On the social level: we want to play soccer and he didn't understand the rules, then they eventually get angry."* |
| **Teachers** | | |
| Intercultural competence | Accepting<br>• cultural differences<br>• religious or cultural practices<br>• differing conceptions of school<br>• differing conceptions of roles and mental health | External 5 *"It is the understanding of the language as well as the understanding of cultural specialities that is what I find essential in contact with these children. That they get the chance to show how specific situations are managed differently in their culture. That is what I find a very important point that we don't presume our normality as for granted for these children."* |
| Teaching style | Providing appropriate learning material | Teacher 4 *"You have to work even more than usual. You always have to have extra material. You have to be even more patient than usual because the class, in general, demands a lot from you. And if you get these children on top of that—actually you cannot do justice to these differences. And that is stressful."* |
| | Empathy | |
| | Understanding | |
| | Patience | |
| Working with parents | Differing gender conceptions | Teacher 2 *"Yes, that was very difficult. Especially with the boys. The boys play a different role in these countries and the girls are more adapted. And they brought that with them here, for sure. They are brought up that way in the family, from their parents, that they are the boss."* |
| | Differing upbringing | |
| | Sexual education | |
| | Promote prejudices | |
| | Establishing a relationship | |
| | Language barriers | |

*(Continued)*

**Table 2.** (Continued)

| Individuals | | |
|---|---|---|
| **Code** | **Example** | **Citation** |
| **Resources** | | |
| Origin is irrelevant | Lessons are not disturbed | External 2 *"You know, in the lesson that is not a topic. Because there, of course, you have help."* |
| | Familiarity with multicultural situations | |
| | Especially young children | |
| | No conflicts | |
| Openness | Interest in other cultures | Student 6 *"Actually he is a really nice person, but when others pick on him then he doesn't like it and then he wants to pick on the others as well. So that it's fair for him. Because he comes from a different country, there is war. He copied them a bit."* |
| | Accepting differences | |
| | Finding commonalities | |
| | Dismantling prejudice | |
| | Comprehension | |
| Enrichment | Getting to know<br>• religions<br>• languages<br>• rituals<br>• food | External 3 *"It begins with how many languages we can sing "Happy Birthday" in. It's true because we always all together learn every language that is spoken in the class across primary school time. But also things like when a new student comes, who for example doesn't speak the language yet, there is definitely often one or two children who can help translate."* |
| Parental integration | Organise parties | Teacher 7 *"If you have a motivated parents' association all parents, for example, help to prepare celebrations in the class or in school. That helps a lot at the basis in dealing with each other without reservations."* |
| | cook food | |
| | talk about their culture | |
| | recognizing similarities | |
| Establish relationships | Children can make friends | External 2 *"Many children like it, you have to say, for them it is obviously cool because they have other children there and that is nice for them."* |
| Helpfulness | Practice empathy | External 2 *"When a child sees that another child doesn't understand something then the first intention of that child is not to say: "You have darker skin than me I won't talk to you", but to say "I will help you". This helpfulness that is in this that is a resource, which can become much more visible there."* |
| | Lend material | |
| | Translate | |
| Preparation for life | Society is colourful | Teacher 5 *"For starters, I think our generation has simply become more colourful over the past years and that is what our children grow up with. I did not have children from other countries in my class when I was in primary school. But school is supposed to prepare for life and that is preparation."* |
| | Accepting everyone | |

The table presents the codes that were extracted from the interviews concerning the consequences of cultural and linguistic diversity for the children themselves, the classmates and the teachers. Every code is explained with examples as well as a citation.

When a child is aggressive, it can be a serious *burden* to the classmates and the entire classroom atmosphere. Some children explained that there were times when they were afraid of going to school. Students also complained about disruptions during lessons that would keep them from concentrating. Teachers, as well as students, observed lower achievements for the entire classroom if one student demands a lot of attention.

## Teachers

The integration of students with mental health issues in the classroom can be a *personal burden* for teachers. Sometimes it inflicts psychological or emotional stress on the teachers which requires the establishment of boundaries. Many teachers talk about feeling left alone. Additional appointments with parents or social workers increase the burden. This can even lead to burn-out. The externals report that between teachers, parents and experts the responsibilities are not clearly distributed. This can cause stress and blame. Externals criticised that teachers lack the training for handling mental health problems which leads to insecurities.

For the teachers, a significant problem is *time management*. As many of them explained they are torn between sticking to the curriculum and trying to discipline the students. Especially when, at the beginning of each lesson, they must deal with conflicts that happened during break-time. This minimises learning time for the entire class.

Another challenge that teachers, as well as externals, recognise is *caring for all students* simultaneously as distressed children can demand a lot of attention. Sometimes teachers try taming one aggressive child while at the same time observing the rest of the classroom. This can make teachers feel like they do not do justice to every student. Besides, teachers are required to train the students' social competencies and integrate every student into the classroom. A few teachers also mentioned situations where they had to protect the class from one aggressive student.

Dealing with mental health issues in students, calls for important *skills* in teachers. They need to be able to reflect on their own behaviour, possibly even with a supervisor. Some students need differential treatment which requires teachers to be able to interpret and classify behaviour correctly. Externals especially mentioned the need for teachers to be patient.

Including students with mental health issues into the classroom makes *collaborating with the parents* or caregivers even more important. This implies additional work for the teachers through house calls and meetings. Some teachers reported situations where parents hindered cooperation when they did not recognise their child's problems. One external raised the concern that there is a lack of transparency between teachers and parents.

**Resources.**   Despite all those the negative effects, mental health issues in children can also lead to *support* from their classmates. The interviewed students showed distinct skills in recognising other's emotions. They explained that they would generally try to help their friends and classmates when they were sad by distracting them or mediating in conflicts. The students displayed an understanding for the behaviour of distressed children even when they were aggressive or loud. These assertions were supported by the teachers.

Externals, students and teachers explained that some students simply *accept* differences between people. They do not judge others and understand the children's behaviour in relation to their experiences. Others focus on their personal goals without letting themselves be distracted by classmates.

## Cultural and linguistic diversity

**Individuals.**   For refugee children or children with a migration background, the biggest barrier in school clearly seems to be the *deficit in the native language*. The lack of language skills leads to learning difficulties and makes it hard for them to concentrate or participate in regular lessons. Besides, conflicts with peers often arise from misunderstandings and the inability to solve a conflict with words. One psychologist that works extensively with children with migration background explained that language deficiencies can lead to low self-esteem which in turn can cause psychological distress or mental health disorders.

Closely related to this issue are insecurities concerning their *social affiliation*. They might neither feel home in the country their parents were born in nor in the country they live in now. Especially children that just arrived in a new country, do not yet have friends that they can rely on and identify with. Teachers observed that children from the same country will often stick together and demarcate from the rest. This is true for immigrated children as well as for children born in the host country.

One teacher mentioned that in her experience some children have more *difficult preconditions*. Especially in wealthy neighbourhoods, immigrated children are constantly confronted with having less equipment than the other children have. Another teacher reported that some refugee children have no schooling experience.

Finally, in some children, you can see the *strain caused by the experiences* they have made in their home country or during their flight. If children experienced traumatic situations it can lead to difficulties in concentrating and in some cases even result in posttraumatic stress disorder. One external explained that some children from troubled countries might not have learned how to solve conflicts without violence.

**Classmates.**    The negative reactions of classmates to children with different cultural or linguistic backgrounds pertain to four categories. If a child acts very different from what children are used to it can lead to *incomprehension* and misunderstandings. Different eating or behavioural habits can be hard to understand for some children if they are used to other customs.

In the worst cases, this incomprehension can cause *exclusion* of certain students. Some classmates use peculiarities to distance themselves from other children or bully them. Teachers, as well as the special needs educators, recount instances of bullying that go in both directions against children with migration background as well as against children born in the host country. One teacher was especially concerned by children with migration background receiving special treatment. This assigns them an outsider position that they often don't want.

Occasionally *differences build a barrier* between children. The topic that was mentioned most by teachers, as well as externals, was religion. Diverging public holidays or values concerning relationships can cause conflicts and insults.

Some teachers recognised that in cultural and linguistic diverse classrooms students have to meet a few *challenging expectations*. They are expected to be patient when a teacher explains something multiple times. Besides, students should be considerate when children do not understand the teacher or receive special treatment. Finally, children are often required to work independently when the teachers focus their attention on single students. All this can be especially challenging for troubled children.

**Teachers.**    When teachers face a cultural and linguistic diverse classroom, they find themselves confronted with new challenges. Teachers need to obtain *intercultural competence* to deal with cultural and religious differences. They must self-reflect on their own normality which might differ from the one of the students or the parents just like concepts of school and mental health.

Teachers need to adjust their *teaching style* when they are confronted with a diverse classroom. The interviewed teachers considered the provision of appropriate learning material as one of the biggest challenges. They are expected to follow the curriculum while simultaneously teaching a new language to some students. Teachers and externals both perceive empathy and patience as crucial skills to strengthen every individual student.

Another challenge that teachers deal with is *working with the parents* of migrant students. Often there is a language barrier, which requires a certain amount of planning before a meeting can take place. Moreover, upbringing in other countries can differ from their own idea, especially when it comes to gender roles. A few teachers and externals stated that cultural or religious conflicts between children do not originate from the students themselves but are attributable to some parents who promote prejudices.

**Resources.**    Despite these challenges, all interview partners were able to identify multiple resources of cultural and linguistic diverse classes. Often the first reaction when it came to diversity was that the *origin is irrelevant* for the children. Especially young children easily find a way to connect and do not care where someone is from. Teachers explained that during lessons the background is not important.

The teachers highlighted the *openness* of their students. Through close contact with diverse cultures, children learn to interact with different people and find commonalities. The teachers see this as a possibility to dismantle prejudices. In the interviews, the children appeared very comprehending of deviant behaviour or differential treatment because of cultural differences.

Teachers and externals consider multicultural classrooms a great *enrichment* because students can contribute first-hand insights on languages, religion, food, or public holidays from other countries. One teacher explained that thereby children learn that differences between cultures are often negligible.

In this context, the teachers stated the commitment and *integration of the parents* as another resource. When parents participate in the preparation of celebrations this collaboration can help to reduce stereotypes.

In addition, the school can be a resource for the children that arrive new to the country. Meeting and studying with peers can facilitate the *establishment of relationships* and enable them to find friends.

Children, teachers and externals all stressed the *helpfulness* of students. The social space manager explained that for most students their initial reaction to a child in need seems to be helping rather than bullying them. One teacher stated that diverse classrooms help *prepare for life* because they reflect our society which is becoming more and more colourful.

## Discussion

The interviews with teachers, externals and students reveal that mental health issues can be a serious burden not only for the affected student but also for classmates and teachers. Cultural and linguistic diversity poses a challenge if language or religious barriers exist. On the other hand, this diversity bears a variety of resources for students, teachers, and parents.

The interviews focused on German primary schools. Primary school classes in the here included cities have between 17–26 students [74]. The distribution of teacher assistants and pedagogues in these schools depends on the amount of special needs students and of students in total in a school and classroom [75]. In most cases, teachers are only supported by an assistant for a few hours a week [76].

The teacher sample was recruited with maximum variation. Yet, only female primary school teachers participated in the interviews. Though the majority of primary school teachers in Germany are female (89.5%) [77] the experiences of male teachers with challenges at school might differ from the ones of female teachers.

Externals are not referred to as experts as no expert interviews were conducted. Expert interviews focus on the expertise of a person and their specific knowledge on a subject [78]. In this study, externals were included to expand the information received from teachers and provide a different perspective on the issue of interest. Therefore, not every question of the interview guidelines pertained to the externals' specific field of expertise.

As recommended in the literature interviews with children should be focussed, short and target specific situations [79, 80]. This limits the possibility of generalising their statements. Nevertheless, it is important to collect the opinions of children, when researching their environment [79, 81].

Even though the teachers and externals worked in diverse locations with varying lengths of experience, and schools with different socioeconomic backgrounds the challenges and problems of mental health issues and cultural and linguistic diversity they recounted were rather similar. Teachers confirmed the evidence that mental health problem can lead to lower achievements in school [4, 5, 16]. In addition, the previously stated risk of being bullied by peers [20, 21] and the difficulty in relationships [6] were often mentioned. Externalising behaviour, especially aggression towards peers, constituted the most dominant challenge in these interviews. Teachers did not talk about grade repetition or leaving school prematurely due to mental health issues [17, 18]. However, this is most likely because only primary school teachers were interviewed.

Some teachers and students verified that symptoms of mental health in one student could lead to lower achievements in the entire classroom [26–28] because teachers have to focus all their attention on single students and lessons are disrupted. Problematic behaviour of one child can incite other students to join in disrupting the class as had been suggested by O'Brennan, Bradshaw [36]. In addition to these ramifications, teachers, students, and externals in this study also explain that mental health issues in children can lead to fear and a deteriorated classroom atmosphere. However, they were also capable of identifying resources such as support and acceptance of differences.

In the literature, the reason why problematic behaviour is stressful for teachers, authors supposed that this is because teachers constantly have to divert their attention from the classroom and discipline the child [82, 83]. Teachers confirmed that one big source of stress is keeping an eye on every student. As previously stated, teachers often feel left alone [31]. Besides, teachers and externals perceive a need to collaborate with the parents which can be challenging at times [31]. Externals also criticised insufficient professional training of teachers in dealing with children with mental health problems [7] as teachers occasionally need to protect students from themselves or others and always interpret their behaviour in relation to the students' history. In addition, teachers and externals explained that the uncertain distribution of responsibilities, the lack of transparency in collaborating with the parents and the additional appointments further add to their stress.

Concerning the cultural and linguistic background of students, the interviews confirm that native language deficits can lead to learning difficulties and concentration problems [8, 9]. However, those were not the only effects mentioned in this study. Deficits in the native language can also lead to problems in conflict communication and lower self-esteem. Similarly, Gray-Little and Hafdahl [47] had identified lower self-esteem in minority students. This study shows that cultural and linguistic differences impede the finding of friends and social affiliation. As previously stated, migration background can lead to bullying by native students [39]. Yet, teachers explained that bullying appears in both directions.

The teachers and externals did not state any concerns regarding the effect of a cultural and linguistic diverse classroom on students' achievement as was done extensively in previous research [8, 42, 84]. Also, a feeling of futility in the native students and a rise in misconduct [46] were not mentioned. While Goldsmith [10] identified an increase in interethnic conflicts, teachers and externals here did not worry about conflicts but more about ethnic grouping, exclusion and barriers that form between students because of incomprehension and challenges imposed on students. Possibly, the results of this study differ from the literature because the effects were visible when comparing different schools, while the teachers here only referred to one school. Besides, teachers, as well as externals, were eager to look at the source of the conflicts instead of simply blaming it on culture.

In line with previous research [11, 52, 57], the results show that it is crucial for teachers to obtain intercultural competence and the ability for self-reflection to be able to teach cultural and linguistic diverse classrooms successfully. Teaching students of varying language abilities was named as very challenging for teachers. Additionally, this study identified the collaboration with parents of different cultural and linguistic backgrounds as important yet complex.

Resources of cultural and linguistic diversity in school had not been a focal interest in research previously. The interviews did, however, uncover a variety of positive effects of diversity in school. Just as it enriches the education at college [67] it can have the same impact on primary school lessons. Besides, it expands the openness and relationships of students. Especially younger children were deemed as very open towards other cultures. This is in line with previous research that found younger native Dutch adolescents to be more in favour of minorities than older adolescents [85].

## Limitations

There are some limitations to this study. No interviews were conducted with children with a migration background. In the classes that were contacted, the language proficiency of the students with a migration or refugee background was not sufficient for an interview. Despite multiple efforts it was not possible to receive parental permission for interviews with students of different cultural and linguistic backgrounds. It would have been interesting to hear their perspective on problems and resources that emerge in cultural and linguistic diverse classrooms. Similarly, no students with mental health issues were interviewed. Due to the protection of data privacy teachers could not disclose information about possible mental health issues in their students. It would be valuable to find out, how children with mental health issues experience their time at school.

The sample size of this study is relatively small. However, despite the great diversity of interviewed teachers and externals there was extensive overlap between their answers so that more interviews would most likely not have produced a lot of new information. Besides, there is evidence that six interviews are sufficient to identify the main themes of a topic [86].

In this sample, the focus was only on primary schools. It is possible that older children perceive other effects of cultural and linguistic diversity or react differently to mental health problems in classmates. Killen and Stangor [87] found that with increasing age adolescents become more interested in social group functioning, issues of social cohesion and society. Therefore, the effects of diversity might be stronger and more adverse in older age groups.

## Conclusion and outlook

Semi-structured interviews with teachers, students and externals were conducted on the challenges that arise through mental health issues or cultural and linguistic diversity in classrooms. Mental health problems reduce the academic as well as the social competence of affected children. Besides, the expression of the symptoms can deteriorate the achievements of the classmates and worsen the classroom atmosphere. Teachers are required to develop skills in dealing with the students and spend extra time on solving conflicts and training social competencies.

Cultural and linguistic diversity may lead to learning difficulties and uncertain social affiliation in students. It can form a barrier between classmates and cause bullying and exclusion. Teachers expressed the need for appropriate material for teaching different languages. Yet, cultural and linguistic diversity bears a variety of resources like facilitating integration, openness and enriching the lessons.

Future research should assess the specific needs of teachers when dealing with mental health problems in students to develop adequate interventions and types of support. Furthermore, older children should be questioned to see how mental health and cultural and linguistic diversity operate in higher classes.

The cultural background of the interviewed teachers was not assessed in this study. However, the culture of a teacher can influence the relationship with their students in various ways. Studies from the United States show for example that teachers rate children of a different ethnicity as more negative [88, 89] and have lower expectations for their educational success [90]. Therefore, assessing cultural background and cultural attitudes of teachers might be an appropriate way to control possible confounders in similar studies.

Mental health issues in children inflict a variety of challenges on teachers and students. Like studies on mental health literacy in teachers [91, 92] and on the teachers' role in dealing with mental health issues in school [30, 34] the results of this study suggest that it would be advisable to include a mental health training into the teacher formation to increase their confidence

and ability in dealing with mental health problems in students. Besides, the effects of cultural and linguistic diversity identified here lead to the assumption that teachers could benefit from an intercultural competence training enabling them to encounter the cultural differences of their students positively and act interculturally appropriate [63, 64].

## Supporting information

**S1 Transcripts. Excerpst of the transcribed interviews with the externals.**
(PDF)

**S2 Transcripts. Excerpts of the transcribed interviews with the teachers.**
(PDF)

**S3 Transcripts. Excerpts of the transcribed interviews with the students.**
(PDF)

## Author Contributions

**Conceptualization:** Sanna Higgen, Mike Mösko.

**Data curation:** Sanna Higgen.

**Formal analysis:** Sanna Higgen.

**Funding acquisition:** Sanna Higgen.

**Investigation:** Sanna Higgen.

**Methodology:** Sanna Higgen, Mike Mösko.

**Project administration:** Sanna Higgen, Mike Mösko.

**Resources:** Sanna Higgen, Mike Mösko.

**Software:** Sanna Higgen.

**Supervision:** Mike Mösko.

**Validation:** Mike Mösko.

**Writing – original draft:** Sanna Higgen.

**Writing – review & editing:** Sanna Higgen, Mike Mösko.

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
