## [Decision Letter · Decision Letter 0]

7 May 2020

PONE-D-20-02232

Mental health and cultural and linguistic diversity as challenges in school? An interview study on the implications for students and teachers.

PLOS ONE

Dear Ms Higgen,

Thank you for submitting your manuscript to PLOS ONE. After careful consideration, we feel that it has merit but does not fully meet PLOS ONE’s publication criteria as it currently stands. Therefore, we invite you to submit a revised version of the manuscript that addresses the points raised during the review process.

We would appreciate receiving your revised manuscript by Jun 21 2020 11:59PM. To enhance the reproducibility of your results, we recommend that if applicable you deposit your laboratory protocols in protocols.io, where a protocol can be assigned its own identifier (DOI) such that it can be cited independently in the future. For instructions see: http://journals.plos.org/plosone/s/submission-guidelines#loc-laboratory-protocols

We look forward to receiving your revised manuscript.

Kind regards,

Vasileios Stavropoulos

Academic Editor

PLOS ONE

Reviewers' comments:

Reviewer's Responses to Questions

**Comments to the Author**

1. Is the manuscript technically sound, and do the data support the conclusions?

Reviewer #1: Yes

Reviewer #2: Partly

2. Has the statistical analysis been performed appropriately and rigorously? 

Reviewer #1: N/A

Reviewer #2: N/A

3. Have the authors made all data underlying the findings in their manuscript fully available?

Reviewer #1: Yes

Reviewer #2: No

4. Is the manuscript presented in an intelligible fashion and written in standard English?

Reviewer #1: Yes

Reviewer #2: Yes

5. Review Comments to the Author

Reviewer #1: The paper overall was a good read and informative. It contained a valuable perspective and tried to investigate a new aspect of the connection between mental health problems and cultural & linguistic diversity and the school environment. The Method and Results section were particularly well written and laid out. The tables with the codes and examples were useful to evaluate the qualitative information and understand what aspects of the categories were being explored. The citations provided an insider account of what the operational and practical explanations of those categories looked like.

The introduction and rationale building up to the research could be more specific. Since cultural diversity is an important aspect of the study, it would be interesting to know what the country of origin is for the research quoted in the literature review. I would also be interested in understanding the magnitude of effect for some of the research quoted (line 65: what is “negatively impacts”?, line 72: “disruptive students can affect the entire classroom”- what does "affect" mean? Where was this research done? What age group is that specific to? Is disruptive in the original study in the context of mental health problems?). I felt that overall the introduction lacked specific information and overlooked some important aspects of the literature that may contribute to a more meaningful discussion.

I would like to know the gender for the experts and the cultural background for the teachers in the classroom. Did the teachers generally identify with the majority culture in the classroom?. Does the relationship across variables and aspects like patience and empathy vary when the teacher is of a minority cultural background?

In the discussion section, it might be important to just mention the typical set up of a classroom since burden and time management skills are of importance when discussing teacher perspective. I would have thought primary classes usually have an assistant teacher? Is that not true for German primary schools? That may have some implications on just the reader understanding the larger picture a little better.

The conclusion and outlook section are very well written and succinct. It would be valuable to elaborate on what intercultural competence (line 513) entails and if there is any reference to relevant studies that can be brought up in the introduction to introduce this idea. Overall, it is an interesting paper that is fun to read and neatly ties multiple variables discussed in the paper.

Reviewer #2: In general the research was a qualitative examination of the diverse effects of mental health issues and cultural and linguistic diversity on students, classmates and teachers yet these two topics were not examined together in any way to identify if they overlapped.

Review Question 1

Recruitment: Although the researchers have indicated purposive sampling was used to select participants/schools its unclear what judgments were used limiting the ability for the results to be tested/repeated. It is unclear how many culturally and linguistically diverse students were at each school and how much experience each teacher had (number of years etc.).

Data collection: one-on-one interviews were used except for one interview in which a teacher and external were questioned simultaneously. This inconsistency is a confound which could have been avoided and the reason for it is not explained. Sample: Although the research was qualitative in its approach the sample size was very small and not representative in the case of primary school teachers. No details were provided on the gender of students selected nor how they were selected. This limited the inferences which the researchers later made in their discussion. Furthermore, despite the topic of interest relating to children with mental health issues and cultural and linguistic diversity, none of the children interviewed fell into these categories.

Discussion: The authors appear to imply that the answers provided by the primary school teachers "are still representative" of general population of primary school teachers despite the lack of representation of male teachers sample and despite the extremely low number of teachers interviewed (n=7). This is an inference contrary to common research/statistical knowledge.

Conclusion and outlook: The authors state that, "It would be advisable to include classes on mental health and intercultural competence into the teacher formation to prepare them for dealing with these challenges," without citing any research (their own findings or otherwise) to support this statement.

Review Question 3

The reviewer could not locate the data underlying the findings

6. PLOS authors have the option to publish the peer review history of their article (what does this mean?). If published, this will include your full peer review and any attached files.

Reviewer #1: No

Reviewer #2: No

---

## [Author Response · Author response to Decision Letter 0]

11 Jun 2020

Dear Reviewers, 

we appreciate that you have taken the time and effort to read our manuscript and thank you for your constructive suggestions. We have addressed your comments in the "Response to Reviewers" and explained how and where we have changed the text. We believe that your suggestions have helped to improve the manuscript.

We would be happy to respond to any further questions or comments you might have.

Kind regards,

Sanna Higgen

---

## [Editor Report · Decision Letter 1]

1 Jul 2020

Mental health and cultural and linguistic diversity as challenges in school? An interview study on the implications for students and teachers.

PONE-D-20-02232R1

Dear Dr. Higgen,

We’re pleased to inform you that your manuscript has been judged scientifically suitable for publication and will be formally accepted for publication once it meets all outstanding technical requirements.

Kind regards,

Vasileios Stavropoulos

Academic Editor

PLOS ONE
---

## [Editor Report · Acceptance letter]

8 Jul 2020

PONE-D-20-02232R1 

Mental health and cultural and linguistic diversity as challenges in school? An interview study on the implications for students and teachers. 

Dear Dr. Higgen:

I'm pleased to inform you that your manuscript has been deemed suitable for publication in PLOS ONE. Congratulations! Your manuscript is now with our production department. 

Kind regards, 

on behalf of

Dr. Vasileios Stavropoulos 

Academic Editor

PLOS ONE